# Using Theory of Change to inform the design of the HIV+D intervention for integrating the management of depression in routine HIV care in Uganda

Joshua Ssebunnya[1,2]*, James Mugisha[3], Richard Mpango[1], Leticia Kyohangirwe[3], Geofrey Taasi[4], Hafsa Ssentongo[5], Pontiano Kaleebu[1], Vikram Patel[6], Eugene Kinyanda[1]

1 Mental Health Research Unit, Medical Research Council/Uganda Virus Research Institute & London School of Hygiene and Tropical Medicine Uganda Research Unit, Entebbe, Uganda, 2 Butabika National Referral Mental Hospital, Kampala, Uganda, 3 Department of Social Work and Social Administration, Kyambogo University, Kampala, Uganda, 4 STD/AIDS Control Program, Ministry of Health, Kampala, Uganda, 5 Mental Health Division, Ministry of Health, Kampala, Uganda, 6 Department of Global Health and Social Medicine, Harvard Medical School, Boston, Massachusetts, United States of America

* joy95h@yahoo.co.uk

**Data Availability Statement:** De-identified data from which this manuscript has been produced will be made available upon request, but the full data set may not be shared due to the qualitative and

## Abstract

There is growing recognition of the burden of depression in people living with HIV/AIDS (PLWHA), associated with negative behavioural and clinical outcomes. Unfortunately, most HIV care providers in sub-Saharan Africa do not routinely provide mental health services to address this problem. This article describes the process of developing a model for integrating the management of depression in HIV care in Uganda. Theory of Change (ToC) methodology was used to guide the process of developing the model. Three successive ToC workshops were held with a multi-disciplinary group of 38 stakeholders within Wakiso district, in the Central region of Uganda. The first 2 workshops were for generating practical ideas for a feasible and acceptable model of integrating the management of depression in HIV care at all levels of care within the district healthcare system; while the third and final workshop was for consensus building. Following meaningful brainstorming and discussions, the stakeholders suggested improved mental wellbeing among PLWHA as the ultimate outcome of the program. This would be preceded by short-term and intermediate outcomes including reduced morbidity among persons with HIV attributable to depression, allocation of more resources towards management of depression, increased help-seeking among depressed PLWHA and more health workers detecting and managing depression. These would be achieved following several interventions undertaken at all levels of care. The participants further identified some indicators of successful implementation such as emphasis of depression management in the district healthcare plans, increased demand for antidepressants etc; as well as various assumptions underlying the intervention. All these were graphically aligned in a causal pathway, leading to a ToC map, contextualizing and summarizing the intervention model. The ToC was a valuable methodology that brought together stakeholders to identify key strategies for development of a comprehensible contextualized

potentially identifiable nature of the raw data (eg, transcripts). The de-identified data meet journal requirements for the minimal data set. Request for data access should be made to the UVRI – REC Chairperson: Mr. Tom Lutalo, tomlutalo@gmail.com and committee member, Mr. Wilber Ssembajjwe, Wilber.ssembajjwe@mrcuganda.org.

**Funding:** This study is an output of the Senior Research Fellowship in Public Health and Tropical Medicine to Eugene Kinyanda entitled, 'Integrating the management of depression into routine HIV care in Uganda (the HIV+D trial)'; grant number 205069/Z/16/Z. The funder had no role in the decision to publish these findings.

**Competing interests:** The authors have declared that no competing interests exist.

intervention model for managing depression within HIV care in Uganda; allowing greater stakeholder engagement and buy-in.

## Introduction

HIV/AIDS continues to be one of the leading global health challenges of our time, with an estimated 37.9 million people living with HIV (PLWHA) globally. Of these, 20.6 million are living in sub-Saharan Africa [1]; an under-resourced region with significant health system constraints [2]. With an estimated 1.4 million people living with HIV/AIDS in Uganda, the central region has been reported to have the second highest HIV prevalence rate at 7.6%, above the national adult prevalence of 5.7% [3]. Studies have shown that people infected with HIV are more likely to develop depression, with approximately 8–50% of persons living with HIV reported to have suffered from depressive disorders (DD) [4–6]. In Uganda, a recent meta-analysis found a pooled depression prevalence of 31% among PLWHA, nearly ten times higher than the prevalence estimates in the general population [7].

Depression in people living with HIV not only affects the quality of life [8], but has been associated with a number of other negative behavioural and clinical outcomes such as more rapid HIV disease progression including mortality [9, 10], poor adherence to HIV treatment, risky sexual behaviour and increased utilization of health facilities [11–13]. In fact, current predictors indicate that both HIV/AIDS and depression will be the first two leading causes of disability globally by 2030 [14, 15].

Although an estimated two thirds of the PLWHA globally are on treatment, the majority of HIV care providers in sub-Saharan Africa do not routinely provide mental health services to address the problem of depression [1]. There is however growing evidence of specific treatments for depression among PLWHA which have shown positive trends in reducing mental health symptoms and HIV disease progression [16–18]. This makes a case for integrating such treatments in routine HIV care [19, 20].

With an estimated 1.4 million people living with HIV/AIDS in Uganda, studies have reported exceptionally high prevalence of depressive symptoms among PLWHA, posing a major challenge in HIV care despite the success in the scale up of anti-retroviral therapy (ART) and consequently increasing mortality [13, 21]. In response to this absence of mental health care in HIV programs, the Uganda National HIV and AIDS Strategic Plan (2015–2020) has called for the integration of mental health and other chronic conditions in HIV care so as to further improve the quality of care and treatment [22]. To operationalize this policy recommendation, the Ministry of Health released guidelines for the treatment of HIV calling for the assessment and management of depression as an integral part of HIV care programs [23]. Although the need for integrating HIV and mental health services is indisputable, the challenges are evident in implementing a service integration model that is cost-effective, and of high quality and impact [24]. Imperatively, such a model should be acceptable and feasible for improving both mental health and HIV treatment outcomes. It is thus imperative that the model undertakes an approach that embraces the perceptions and experiences of multi-disciplinary groups of stakeholders.

It is against this background that the HIV+D program employed the Theory of Change (ToC) methodology to develop and evaluate an intervention model for integrating the management of depression in HIV care in Uganda. This work was undertaken a priori, working with a programme 'to be' while it is being designed. The orthodox view is that a theory of

change would generally be developed a priori, guiding the intervention design as it is underway and before a programme's implementation model is fully determined, and then revisited at various junctures over time.

A theory of change is in essence a planned route to outcomes, describing the logic, principles and assumptions that connect what an intervention or programme does, how and why it does it, with its intended results [25]. The approach provides for wide stakeholder participation in developing interventions that are contextually appropriate [26]. There are a few published reports on the use of ToC guidelines in programme design, especially complex health interventions. For example, it was used in the program for improving mental health care (PRIME), a multi-country complex intervention aimed at generating evidence on how to integrate mental health into primary care through the development, implementation and evaluation of district level mental health care plans for priority disorders [26, 27].

The HIV+D program is a 5-year project implemented by the Mental Health Section of the Medical Research Council (MRC)/Uganda Virus Research Institute (UVRI)& London School of Hygiene and Tropical Medicine (LSHTM) Uganda Research Unit, in partnership with the STD/AIDS Control Program of the Ministry of Health, Uganda. The project is in response to the 2016 Ministry of Health policy initiative and guidelines which call for the assessment and management of depression in PLWHA [22]; implemented in 3 districts within Uganda.

In this paper, we describe how the Theory of Change (ToC) methodology was used during the formative phase of the HIV+D program to guide the development of the intervention for integrating the management of depression in HIV care in Uganda.

## Methods

The Theory of Change (ToC) approach was developed by the Roundtable on Community Change (Aspen Institute, USA) to evaluate complex community-based change interventions; and seeks to establish the links between intervention, context and outcome [28–30]. The approach was thus used in this study to describe how the HIV+D intervention would work and the intended results. This was in light of the fact that it increases the likelihood that stakeholders will clearly specify the program's intended outcomes, the activities that need to be implemented in order to achieve those outcomes, and the contextual factors that are likely to influence them.

### Context

This component of the study was conducted in Mpigi district as part of the formative phase for developing and piloting the intervention model. This is a peri-urban district, with the headquarters about 25 kilometres from the capital city. The district has socio-demographic and health indicators similar to those of most other districts in the country. For example, the population is predominantly semi-literate (80%); and mostly involved in subsistence farming. Fifty five (55%) of the population is aged below 18 years [31]. The district has one of the highest HIV prevalence rates in the country at 8.0%; above the national average of 5.3% [32].

The district has a well-established and facilitated health service network comprising of 41 health facilities. At the apex is a private-not-for-profit general hospital, which also serves as the referral centre. Below this are Health Centres IV, III, II and I. Health Centre I, which is the lowest level comprises of Village Health Teams (VHTs) or Community Health Workers who are individual health volunteers, often expert clients who may or may not be formally trained who link the community to the formal health service. HIV care is formally provided from Health Centre IIIs to hospital level, with varying range of patient loads. HIV specialized clinics are mostly ran at facilities at the level of Health Centre IV and the district general hospital.

However, there is currently no mental health care in HIV care services provided at the public health facilities. This is partly attributed to shortage in staffing levels (both mental health professionals and primary care staff), low demand for formal mental health services as well as reluctance of the primary care providers to engage in mental health care [33, 34].

## Study design

This was a qualitative exploratory study, based on workshops as the primary method for data collection. The design was considered ideal as there are no previous studies in Uganda that have used ToC as a tool to guide planning and delivery of integrated mental health and HIV services.

## The ToC process

The ToC process hinges upon defining the necessary preconditions required to bring about a given long term outcome. The stakeholders identify the long-term goal (impact of the programme) and think in backward steps to identify the intermediate changes that would be required to cause the desired change, thereby creating a set of connected outcomes (pathway of change). The outcomes are operationalized by identifying indicators which will determine whether the outcome has been achieved. In addition, the evidence base or rationale of how one outcome leads to the next is articulated and the interventions/preconditions required to achieve this. The participants also articulate several assumptions about the change process, and decide a ceiling of accountability where the programme is no longer directly responsible for the outcomes achieved. The ultimate ToC should be plausible, do-able and testable and able to be represented graphically in a ToC map [35, 36].

## Participants and their selection

Selection of the stakeholders to participate in the process was purposive. They were a diverse sample identified and invited by the District Health Officer and the HIV focal person in the district; which gave the process local legitimacy, while capturing the required expertise. They included district health service managers, members from the political leadership and administrative officers, health facility managers, primary health care service providers, district HIV focal person, staff from NGOs involved in health, mental health specialists, religious leaders, Community Health Workers and HIV care service users/expert clients. There are quite a number of people who hold positions in each of these groups/categories, and the privacy of the participants cannot be breached. The process sought to develop a model which would not be deemed cumbersome in terms of resource needs for the implementation. Furthermore, political buy-in was considered crucial for the success of the program. It was thus imperative that the district political leadership was involved in the process.

## The ToC workshops

The workshops aimed at developing a ToC map, also viewed as a visual map reflecting the structure of the HIV+D intervention, and contextualizing the intervention. A total of three (3) ToC workshops were held with a range of stakeholders over a 4-month period (between November 2017 and February 2018). The workshops focused on 5 key areas that pertain to the intervention, including intervention activities, outcomes, indicators, assumptions, and impact. The workshops were facilitated by the first author (JS), assisted by JM as a co-facilitator and were all conducted in a spacious venue at the district headquarters. Cautious of the possible impact of power differential, the participants were split into two groups.

The first workshop had a total of 18participants, mostly at managerial level. These included representatives from the District Council, members from the District Health Management Team, representative from other sectors and NGOs involved in health as well as expert clients (PLWHA). The workshop was conducted a few weeks after the official launch of the project in the district. The workshop began with the Principal Investigator (EK) making a general presentation on depression among PLWHA, highlighting the need to integrate management of depression in HIV care. The Project Coordinator (RM) would then make a summarized presentation highlighting the project outline. The lead facilitator for the ToC then introduced the concept of ToC as 'a system of ideas intended to explain how we think change will happen in the area we want to address, and how we intend to work to influence these changes'; further explaining the key elements of a ToC map. After setting the scene, a discussion followed, during which participants gave their views based on their expertise and experience. This began with participants identifying the intended impact of the HIV+D program. The brainstorming continued with participants working backwards towards identifying various key outcomes and other preconditions for the program, as described earlier under the ToC process. There was note-taking in order to record information from all stakeholders. In addition, the discussions were audio-recorded and later transcribed verbatim, to ensure that all relevant data is captured. The scope of the discussion considered 3 levels of health care at district level, including health care management level, health facility level and community level.

The second workshop had 20 participants including frontline health workers, religious leaders, community health workers and HIV service users/expert clients. A similar procedure (as for workshop 1) was followed. The research team then held internal meetings to synthesize and analyze the views generated from both ToC workshops. A draft ToC map was consequently developed as a major output of the process clearly specifying the causal pathways leading to the desired impact of the program. The map highlighted the activities/interventions to undertake, indicators, assumptions and the expected outcomes.

The third workshop brought together29 participants who had been involved in the first 2 workshops, and a technical officer from the HIV control program at the Ministry of Health headquarters, making a total of 30 participants. The research team presented findings from the previous workshops and the draft ToC map, which formed the basis for a further discussion and review of the draft ToC map until there was consensus towards refinement of the ToC map. Thus, the first 2 workshops were for brainstorming and generating ideas from different groups, while the third workshop was largely for review of the draft ToC map and consensus building.

All the workshops were conducted in the district headquarters council hall; each being a day-long workshop, with participation as summarized in Table 1 below.

## Data collection

The proceeding of each workshop was audio-recorded so as to capture all relevant contributions and suggestions from the participants. As the transcribing of recordings is itself an interpretive process, it was undertaken by the ToC workshop facilitator, assisted by another member of the research team.

## Data analysis

After the transcription, conventional content analysis was done by the 2 workshop facilitators who have expertise in qualitative research and data analysis, thereby ensuring rigour. They subjectively interpreted the content of the transcripts through coding and identifying themes [37, 38]. The initial coding of the transcripts was undertaken by the first author, with the

**Table 1. Participants per ToC workshop.**

| Participants | ToC workshops | | |
|---|---|---|---|
| | *Workshop 1* | *Workshop 2* | *Workshop 3* |
| District Council representatives | 2 | - | 2 |
| District Health Management Team | 8 | - | 7 |
| Representatives from other sectors | 2 | - | 2 |
| Technical Officer, HIV control program, MoH | - | - | 1 |
| Mental health specialists | 1 | 1 | 2 |
| Frontline PHC workers | - | 8 | 6 |
| Religious leaders | 1 | 2 | 2 |
| Staff of NGOs involved in health | 2 | - | 1 |
| Community Health Workers | - | 4 | 2 |
| PLWHA/Expert clients | 2 | 5 | 5 |

coding categories directly derived from the content of transcripts, without imposing any pre-conceived theoretical perspectives. The key components of the ToC (outcomes, indicators, assumptions and activities) were treated as the main categories that served as themes and sub-themes, to guide the coding process. The coded transcripts were shared with the first co-author for common interpretation, improvement and consensus. We focused on what the informants actually said and consequently elicited meaning and valid inferences under each category, to feed into the ToC causal pathway.

## Ethical considerations

Ethical approval was obtained from the UVRI Research and Ethics Committee, the LSHTM Ethics Committee and the Uganda National Council for Science and Technology (ethical clearance number: HS645ES). All stakeholders who participated in the ToC process gave written informed consent to participate.

## Results and discussion

The ToC workshops aimed at generating ideas to inform the development of the HIV+ D intervention and engaging key stakeholders to ensure buy-in and support for the programme. The stakeholders' views and suggestions were synthesized and summarized under the different core elements of a ToC, as presented in the summary narrative below:

### a) Impact

The stakeholders agreed on the overall impact of the intervention being improved mental well-being among HIV patients. This was based on the expectation that the interventions undertaken under the programme would circumstantially address any other probable co-occurring common mental disorders in addition to depression.

### b) Outcomes

The participants mapped out several outcomes (both long term and short term) that would have to be realized and eventually lead to the above desired impact. The long term outcomes included: having depression managed alongside HIV care, reduced morbidity attributable to depression in HIV, reduction in HIV related stigma and better clinical outcomes for HIV patients. These would follow the intermediate and short-term outcomes such as allocation of more resources for depression management following increased political buy-in and support

for the programme; reduced incidence and prevalence of depression in HIV patients once patients with HIV and depression receive appropriate treatment. This would be preceded by health workers handling HIV patients being able to detect and treat depression, appreciation of mental health issues in HIV care and having more psychiatric nurses involved in HIV care. Other short-term outcomes include increased help-seeking for depression among HIV patients and empowerment of persons living with HIV. These would be preceded by increased and demand for services.

They asserted that managing depression would promote the quality of life for persons living with HIV/AIDS and promote their mental wellbeing; and hence the need to routinely assess for depression among all PLWHA.

"...*We want to promote wellbeing because what we have now is quality of life. So one of the things, it takes away their mental wellbeing... because when you manage depression you have managed anxiety as well because we are looking at the common mental disorders. We want to promote mental wellbeing*" (Health facility manager)

Another participant added:

"...*we are talking of assessment. We need the health workers to understand the signs and symptoms of depression so as to be able to identify those patients easily. Otherwise, they will keep missing. As you know, in our culture, there is no word like depression*" (Health Educator)

## c) Interventions

The participants identified several interventions/activities to be undertaken at the different levels so as to realize the outcomes.

At the health management level, there was need for sensitization of the district leadership and implementing partners on the need to integrate mental health care in HIV care especially to ensure support in terms of resource allocation and buy-in, building capacity for monitoring, supervision and follow up as well as strengthening the referral system.

At the health facility level where the bulk of work is expected, the activities to undertake include training of the PHC workers in mental health care, with particular emphasis on depression management; availing health facilities with screening tools for depression (PHQ-2 and PHQ-9); mandatory screening and assessment for depression at the triage desk, availing a start-up package of anti-depressants to support the program, building capacity of health workers to offer psychotherapy, involving psychiatric nurses in HIV care as well as regular monitoring and support supervision.

At the community level, activities include sensitization of expert clients and support groups of PLWHA, sensitizing the communities and emphasizing the management of depression during the health talks, training the community health workers to be able to screen for depression in the community as well as monitoring and follow up of the clients in the community.

## d) Indicators

The participants further identified a number of indicators to measure both short-term and long term outcomes, and tell whether the intervention is on course. First, having the management of depression in HIV care emphasized in the district health plans was identified as an indicator to measure political buy-in and support for the programme as well as allocation of

resources towards management of depression. An increase in disclosure and positive living would be an indicator that depression is managed alongside HIV, consequently leading to better clinical outcomes for patients with HIV. An increase in demand for anti-depressant medicines at the health facilities would signify an increase in the number of patients receiving appropriate treatment for depression; as would be reflected in the Health Management Information System (HMIS). Similarly, increase in the number of patients recovering from depression, as reflected in clinical audit reports would be an indicator of reduced morbidity attributable to depression among PLWHA. Furthermore, an increase in the number of patients receiving treatment and the number of referrals would signify improvement in the help-seeking and case detection for depression in PLWHA.

### e) Assumptions

According to the participants, implementation of the activities to lead to the desired outcomes would be successful on the assumption that there is:

i. political buy-in and support towards the program

ii. supportive policies and plans in place

iii. an efficient Health Management Information System (HMIS), with the district Bio-statistician active and involved

iv. effective monitoring and supervision by the District Health Management Team

v. the required medicines and logistical supplies are in place

vi. competent health workers with a positive attitude

vii. an efficient HIV care system

All these core elements were conceptualized and summarized in a causal pathway, to produce a Theory of Change map that conceptualizes the HIV+D program presented in Fig 1 below.

The stakeholders reported that because of poor mental health literacy in the community including among PLWHA, there was generally low recognition of depression as an illness that is treatable and hence poor help seeking behavior, which negatively impacted the prognosis of their HIV/AIDS illness. The PLWHA on the other hand expressed their reluctance to engage with Community Health Workers due to the latter's weakness in maintaining confidentiality. These were believed to be potential threats to successful implementation of the program. They believed that persons with lived experience (HIV expert clients), were a better alternative as they tend to be more reliable and efficient in providing the much needed peer support. Their availability and involvement was thus considered an opportunity.

The stakeholders further reported that most PLWHA have numerous mental health problems, which often go undetected and unattended to, due to limited awareness and inadequate mental health services. The ToC process identified mentoring, monitoring and supervision of the health workers as crucial inputs for the intervention; and this was considered to be a direct oversight role of the district health managers and the psychiatric nurses.

They further argued that depression among PLWHA can be attributed to several factors including the fear of the disease itself, the associated stigma and several others related factors; mostly co-occuring with some other mental health problems. They thus suggested a need for capacity building in mental health care to enable the service providers understand the likely causes and intervene appropriately.

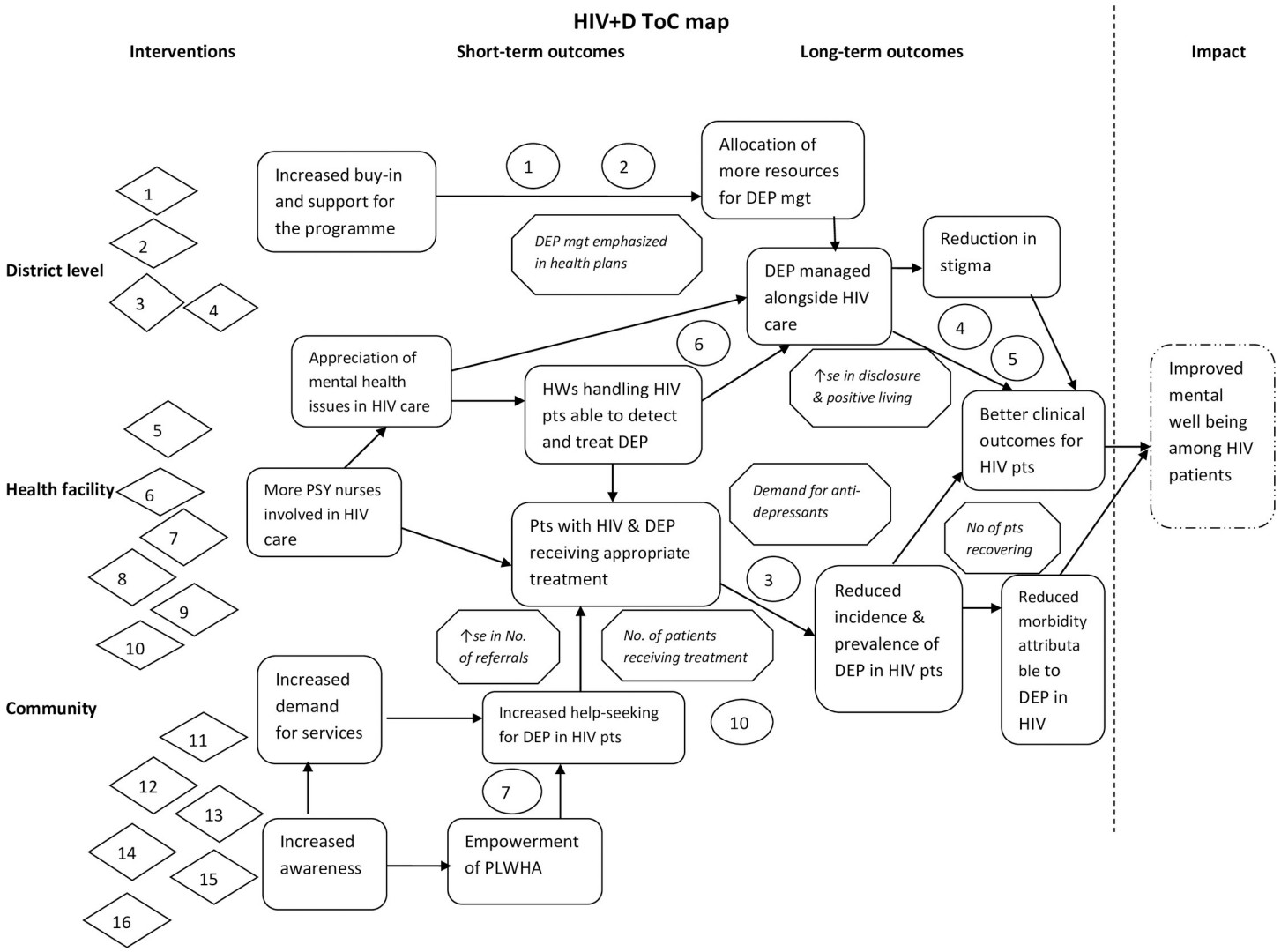

**Fig 1. The HIV+D ToC map.**

"...I don't know whether it is a concern. But in your beginning, you talked about identifying the barriers and then eventually we come up with the strategies–you drew that picture am seeing you as someone who has stopped at the strategies. What about the "why?", the why question. How are we answering it. Like we can ask ourselves what brings depression...||...we might think the depression is because of HIV, but there might be certain other factors that are contributing to it"(District Health Officer)

The health workers in particular emphasized the need for building their competence, citing the challenges in detection and management of depression.

"...you see, this depression in most cases is a feeling inside a person that you are not able to see. No patient will come and tell you that they are depressed. So, it requires some level of competence to be able to detect the ones that are depressed" (General Nurse)

Some of the participants expressed concern over inadequate staffing and resources at the health facilities, affecting the efficiency and service delivery.

*If am a patient, I will want go where I can find drugs. So I want it to be a functional health . . . where one can find all the drugs, the health workers, the health providers and other things. But sometimes, this is not the case (Representative from other sector)*

Another participant added:

*The understaffing is serious. The triage is supposed to be done by a nurse; but at our place, it is us [expert clients] who do it. That is the reality (frontline health worker)*

Furthermore, they believed the planned shift to community health worker-led antiretroviral therapy delivery (CLAD) would mean more responsibility and engagement of the community health workers and expert clients. In light of this, they recommended that it would be imperative for the program to build the capacity of community health workers and peer-support providers to improve case detection and provision of appropriate support.

The participants further noted that under a decentralized system of service delivery, districts have several implementing partners and resources to leverage and promote this integration, making it important to have political buy-in and support. They thus emphasized the need for health system strengthening to ensure support and sustainability, not only at policy level but also in practice.

*"By and large we need a system that will support what we have said the removal of depression. How do we want that system to look like-that system that will support the removal of. . . we need a system that will support those processes. We need the medicines, infrastructure in place, tools and job aids—that complex thing" (District Health Officer)*

## Discussion

The article describes how ToC workshops were used to develop a feasible intervention model for integrating the management of depression in HIV care in Uganda; a policy recommendation by the Ministry of Health in the National HIV/AIDS Strategic Plan (2015–2020). The approach was useful for this purpose, and has been recommended by several researchers [26, 39, 40]. The process resulted in a clear visual map conceptualizing the intervention at all levels of health care within the district system. Furthermore, the process enabled stakeholders to appreciate what has to be done at each level, and why.

This component of the study was part of the formative work, for developing the HIV+D intervention programme, which would later be rolled out and evaluated in a trial, to be conducted in three districts.

The participatory nature of ToC approach created an opportunity for stakeholders with varying expertise and levels of seniority at different levels to brainstorm and discuss issues pertaining to mental health in HIV care from different perspectives. The direct involvement of stakeholders from the district political administration, the deputy accounting officer, the District Health Officer (overall health manager in the district) and a technical officer from the AIDS Control Programme at the Ministry of Health added weight to the process, making it more legitimate. As a result, it was possible to obtain stakeholder buy-in, thereby increasing the chances successful implementation. By having separate workshops for stakeholders based on level of seniority, we avoided power relationships and the likely hindrances, thereby there having increased participation and enabling all participants to give freely their views.

It is important to note that Uganda's health system is highly decentralized, with most of the PHC services availed in public health facilities under the district health system [41, 42]. By

developing interventions at that level, there is a sense of ownership and prioritization, which often determines the likelihood of implementation.

The need to engage HIV expert clients and members of the Village Health Team in the programme activities was emphasized. Use of the expert clients was particularly considered a strength, given their experience and track record of being motivated to work as volunteers for the plight of fellow patients. The expert clients however expressed concern as regards the VHT members and their ability to maintain confidentiality; which could probably be addressed through adequate training and supervision.

Nearly all participants were in concurrence with several earlier studies [4–7], asserting that depression is common among PLWHA. They however expressed concern over the irregular supply of psychotropic medicines and frequent stock-outs, to which it was suggested that start-up packs of antidepressant medicines be availed at health facilities. Furthermore, the main therapy for depression being advocated in the HIV+D intervention was behavioural activation based psychotherapy and not medication and hence would not suffer from limitations related to medicines supply.

The participants also identified a need to improve health worker competences and skills in managing depression through training, mentorship and supervision. However, in light of the evidence that in-service training alone may have limited impact on the practice of health care [43], improvement on competence needs to focus on more than just training, including attitude change as well as close monitoring and supervision. This aspect of capacity building would also call for development of guidelines for the PHC workers as well as the community health workers. Furthermore, they recommended building capacity of health workers to offer psychological treatments. This is in support of the WHO recommendation of the use of psychological interventions as first-line treatment for depression in low income and middle-income countries. A related study by Nakimuli et al. [44] found the integration of group support psychotherapy by trained lay health workers in routine HIV care effective in the treatment of mild to moderate major depression among PLWHA.

Development of this model intervention was timely to support the operationalization of the recommendation by the Ministry of Health to integrate the management of depression in HIV care. At the end of the exercise, the ToC participants expressed confidence that the intervention would positively impact on the care and mental well-being of PLWHA if the resources required for the implementation are availed. There are also lessons to learn from this ToC process by both the district local government and Uganda Ministry of Health.

## Limitations

One major limitation was the number of workshops conducted. Although the total number of participants (38) may be deemed adequate and a representative sample of stakeholders involved in mental health and HIV care in the district, only 3 workshops were conducted in light of the busy schedule of most participants. However, there was adequate time for all stakeholders to actively participate, and enrich the discussions given that these were day-long workshops.

Despite the limitation, one important key lesson was the importance of separating participants by level of seniority and its impact on their active participation. Furthermore, the intervention revolves around use of the existing resources (in light of the limited resources for health) which should ideally facilitate the implementation.

## Conclusion

Given its participatory nature, the ToC approach proved valuable in developing and contextualizing a comprehensible intervention model for integrating the management of depression

within the HIV care system in the district, which was considered feasible and acceptable. Importantly, meaningful involvement of the district health managers, created ownership, buy-in and support for the programme. The study thus generated knowledge on the practicality of how to integrate the management of depression in HIV care, a policy recommendation, backed by several studies locally and globally.

## Supporting information

**S1 Checklist. COREQ (COnsolidated criteria for REporting Qualitative research) check-list.**
(PDF)

**S1 File. Proceedings of ToC workshops.**
(DOCX)

**S2 File. Proceedings of ToC workshops.**
(DOCX)

**S3 File. Proceedings of ToC workshops.**
(DOC)

## Author Contributions

**Conceptualization:** Joshua Ssebunnya, Vikram Patel, Eugene Kinyanda.

**Formal analysis:** Joshua Ssebunnya, James Mugisha, Leticia Kyohangirwe.

**Funding acquisition:** Pontiano Kaleebu, Vikram Patel, Eugene Kinyanda.

**Investigation:** Joshua Ssebunnya, Richard Mpango, Hafsa Ssentongo, Eugene Kinyanda.

**Methodology:** Joshua Ssebunnya, Hafsa Ssentongo, Vikram Patel, Eugene Kinyanda.

**Project administration:** Richard Mpango, Eugene Kinyanda.

**Supervision:** Geofrey Taasi, Hafsa Ssentongo.

**Validation:** Leticia Kyohangirwe, Geofrey Taasi.

**Writing – original draft:** Joshua Ssebunnya.

**Writing – review & editing:** James Mugisha, Richard Mpango, Geofrey Taasi, Pontiano Kaleebu, Vikram Patel, Eugene Kinyanda.

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
