## [Decision Letter · Decision Letter 0]

17 Mar 2021

PONE-D-21-03331

Using Theory of Change to inform the design of the HIV+D intervention for integrating the management of depression in routine HIV care in Uganda.

PLOS ONE

Dear Dr. Ssebunnya,

Thank you for submitting your manuscript to PLOS ONE. After careful consideration, we feel that it has merit but does not fully meet PLOS ONE’s publication criteria as it currently stands. Therefore, we invite you to submit a revised version of the manuscript that addresses the points raised during the review process.

We look forward to receiving your revised manuscript.

Kind regards,

Amrita Daftary

Academic Editor

PLOS ONE

Journal Requirements:

2. Please include your tables as part of your main manuscript and remove the individual files. Please note that supplementary tables should be uploaded as separate "supporting information" files.

5. Please ensure that you refer to Figure 1 in your text as, if accepted, production will need this reference to link the reader to the figure.

6. Please include a caption for figure 1.

7. Please upload a copy of Figure 2, to which you refer in your text on page 16. If the figure is no longer to be included as part of the submission please remove all reference to it within the text.

Reviewers' comments:

Reviewer's Responses to Questions

**Comments to the Author**

1. Is the manuscript technically sound, and do the data support the conclusions?

Reviewer #1: Partly

Reviewer #2: Partly

2. Has the statistical analysis been performed appropriately and rigorously? 

Reviewer #1: N/A

Reviewer #2: N/A

3. Have the authors made all data underlying the findings in their manuscript fully available?

Reviewer #1: Yes

Reviewer #2: Yes

4. Is the manuscript presented in an intelligible fashion and written in standard English?

Reviewer #1: Yes

Reviewer #2: No

5. Review Comments to the Author

Reviewer #1: This is an interesting manuscript that describes how the Theory of Change was used to build consensus, identify key challenges in programmatic development and service delivery, and to develop a consensus plan for potential implementation of programs to treat comorbid depression in PLHIV. The manuscript is well-written, and the authors appropriately dedicate a considerable amount of space describing the Theory of Change as a process for developing strategies to address public health problems. Mental health is a particularly difficult area to work on in LMICs, in light of the lack of dedicated personnel in most contexts, so this process was particularly helpful in highlighting the problem to key stakeholders and building a plant to address it. The manuscript is well-written.

I have only one major comment with regard to the Results section - which could have meaningful implications for re-writing that section and making it more robust. The authors have noted that the TOC focus groups that were conducted with stakeholders were audio-recorded and that transcripts were created of these recordings. However, only one representative quotation is included in the results section on page 17 (describing challenges in detecting depression in these patients).

In comparison to the rest of the paper, the Results section seems rather thin, and it would benefit from better presentation of empirical data that informed the TOC map / figure, which is the main finding of their research process. However, all of the critical points in this TOC map were presumably informed by actual statements and feedback provided in the stakeholder meetings / focus groups. The authors should strongly consider including selective representative quotations on key points raised by stakeholders, with regard to their concerns about resources limitations, the need for higher level buy-in and resource provision, challenges raised by community stakeholders, etc. This will not only make the paper more rich, but it will help empirically justify and clarify what is in the TOC map - which otherwise seems to arise from thin air.

Obviously, there are limited quotations that can be introduced, but at least 5 or 6 should be considered to flesh out key points in the TOC map.

Reviewer #2: Well done Authors.

Please find areas to be attended to below.

Abstract

Method: Please name the district, region and country where the study was conducted.

Please maintain either PLWH or PLWHA in all write-up instead of interchanging them.

Background

PLWH prevalence in Uganda and if possible the region and district of study site? This should appear in paragraph 1. The rates in Uganda/region/district of study may be lower than the global and sub-Saharan but these rates could be increasing or not changing over a range of years and this may also be a concern I think. What about the association with depression in Uganda, region and district of study site?

Methods

When was this study conducted?

How did you take care of anonymity and confidentiality?

Ethical

Ethics clearance number for UNCST not quoted.

Data Analysis in 2 sections?

Please merge the different sections to have only one on data analysis. The first section seems hanging.

The merging should be well labelled as 2 different sections namely:

Data Collection

Data Analysis

Work Published elsewhere?

This article should be referenced so that readers can refer to it. In case it is not yet published, then the reader will be disadvantaged.

Results

The study had a number of individual and group discussions but in this section, only one conversation has been quoted? Each subsection (of impact, outcomes, interventions, indicators) should be accompanied by at least 1 conversation, otherwise if left plain as it is makes the reader think the authors have written their perceptions.

Discussion

Are there any of your findings which other studies did find too?

What new knowledge has your study added to the field of research and science?

Conclusion

This seems so thin. Of what importance are the study findings to individual patients, health decision makers and implementers, Uganda and the globe if any?

Authors’ Contribution

RM is the project lead or RM as the project lead…... Can one of words “the” be deleted so the statement end as …in the study design?

References

Can references with internet links be accompanied with when this was accessed?

There is need for edits to be done on the references. For some, the journal is in italics and others not. Is the journal name and year of publication separated by a dot or not? Is year, journal number and pages supposed to have spaces or not? There has to be uniformity.

6. PLOS authors have the option to publish the peer review history of their article (what does this mean?). If published, this will include your full peer review and any attached files.

Reviewer #1: **Yes: **Ramnath Subbaraman

Reviewer #2: No

---

## [Author Response · Author response to Decision Letter 0]

18 Apr 2021

The attached documents fully address the specific editor and reviewer comments.

---

## [Editor Report · Decision Letter 1]

24 May 2021

PONE-D-21-03331R1

Using Theory of Change to inform the design of the HIV+D intervention for integrating the management of depression in routine HIV care in Uganda.

PLOS ONE

Dear Dr. Ssebunnya,

Thank you for submitting your manuscript to PLOS ONE. After careful consideration, we feel that it has merit but does not fully meet PLOS ONE’s publication criteria as it currently stands. Therefore, we invite you to submit a revised version of the manuscript that addresses the points raised during the review process.

Thank you for responding to the reviewers' comments. However there are a few issues relate to reporting qualitative studies that needs to be addressed further:

1. Please review and adhere to the journal's expectation for reporting qualitative research: https://journals.plos.org/plosone/s/submission-guidelines#loc-qualitative-research

2. There is very limited information given around the data analysis. No references are provided for content analysis. The subsequent interviews and analysis which proceeds as part of another phase of this study need not be mentioned here at all. However far greater information about the analysis process for this set of workshops is needed. Simply stating content analysis is in adequate. Who analyzed the data, were software used, how was rigour facilitated, how was the ToC used to guide analysis / themes, etc. (Completing a checklist such as COREQ may assist the team in building up their analysis section.)

3. The supplementary files are raw data, including a lot of information about participants, especially page 1 of each supplementary file (and upon reading some of the transcripts, potentially identifiable data about the participants, by way of their descriptions and stories). Whether this level of sharing outside of the study team was included in the consent form is not clear. So the authors should either share the consent form to clarify what exactly participants agreed to, or then please remove all supplementary transcripts and state that de-identified data will be made available upon request, but the full data set may not be shared due to the qualitative and potentially identifiable nature of the raw data (eg, transcripts).  

4. Similarly, Table 1 must be removed as it may be too revealing. I gather there are only a few people holding some of these positions in Mpigi district and it is unclear if their privacy could be breached. If this is not the case, then the authors may retain Table 1 but make it clear in the text that many people hold these positions, to ensure no one participant's identity is inadvertently breached (eg, MH specialist, Asst Chief Admin Officer). I suggest stating that 1) 3 workshops were held with a range of relevant stakeholders - including the various stakeholder/participant types (add total N and n per workshop), and 2) that workshops included people working in similar job ranks to enable open dialogue, etc. If there is anything further the viewer should know about the workshops, for example, key differences between the three workshops in terms of questions asked/data collected or the way the interviews went, then please share that as well. It is for example unclear if the same people attended more than one workshop or all participants were unique. If same people attended >1 workshop , then we need to understand exactly what the differences between the workshops were (eg, in terms of questions asked).

We look forward to receiving your revised manuscript.

Kind regards,

Amrita Daftary

Academic Editor

PLOS ONE

Journal Requirements:

Additional Editor Comments (if provided):

Please see points 1-4 made above. Thank you.

---

## [Author Response · Author response to Decision Letter 1]

28 Sep 2021

We have addressed the recent comments by the Editor

---

## [Editor Report · Decision Letter 2]

20 Oct 2021

Using Theory of Change to inform the design of the HIV+D intervention for integrating the management of depression in routine HIV care in Uganda.

PONE-D-21-03331R2

Dear Dr. Ssebunnya,

We’re pleased to inform you that your manuscript has been judged scientifically suitable for publication and will be formally accepted for publication once it meets all outstanding technical requirements.

Kind regards,

Amrita Daftary

Academic Editor

PLOS ONE
---

## [Editor Report · Acceptance letter]

17 Nov 2021

PONE-D-21-03331R2 

Using Theory of Change to inform the design of the HIV+D intervention for integrating the management of depression in routine HIV care in Uganda. 

Dear Dr. Ssebunnya:

I'm pleased to inform you that your manuscript has been deemed suitable for publication in PLOS ONE. Congratulations! Your manuscript is now with our production department. 

Kind regards, 

on behalf of

Dr. Amrita Daftary 

Academic Editor

PLOS ONE